# Redundant Photo-Voltaic Power Cell in a Highly Reliable System

**Bertalan Beszédes \*** , **Károly Széll** and **György Györök**

Alba Regia Technical Faculty, Óbuda University, H-8000 Székesfehérvár, Hungary;
szell.karoly@amk.uni-obuda.hu (K.S.); gyorok.gyorgy@amk.uni-obuda.hu (G.G.)
**\*** Correspondence: beszedes.bertalan@amk.uni-obuda.hu

**Abstract:** The conversion of solar energy into electricity makes it possible to generate a power resource at the relevant location, independent of the availability of the electrical network. The application of the technology greatly facilitates the supply of electricity to objects that, due to their location, cannot be connected to the electrical network. Typical areas of use are nature reserves, game management areas, large-scale agricultural areas, large-scale livestock areas, industrial pipeline routes, water resources far from infrastructure, etc. The protection of such areas and assets and the detection of their functionality are of particular importance, sectors classified as critical infrastructure are of paramount importance. This article aims to show the conceptual structure of a possible design of a high-reliability, redundant, modular, self-monitoring, microcontroller-controlled system that can be used in the outlined areas.

**Keywords:** redundant; robust; self-monitoring; embedded system; modular PSU; high reliable; solar energy; off-grid; object protection

## 1. Introduction

Despite the arrival of many renewable energy sources, the vast majority of the world's energy supply is still provided by fossil fuels, the extraction and use of which involves the release of large amounts of greenhouse gases into the atmosphere. Therefore, improving energy efficiency is currently one of the most effective means of trying to fight climate change. While the world's energy efficiency is improving [1], the amount of wasted energy is decreasing. As a result of the growth of Earth's population and global economy, humanity's total energy consumption (global primary energy demand) continues to rise. Additionally, the predominance of fossil fuels is leading to an increase in greenhouse gas emissions, which have recently grown at their fastest pace since 2013.

Technological advances have made it possible to increase energy efficiency, which significantly reduces emissions while increasing energy use. Energy efficiency could be improved at a much higher rate, with technologies that are already available, but rarely used [2–5].

According to the International Energy Agency (IEA), the potential is enormous, and by taking advantage of it alone, we could stop the rise in greenhouse gas emissions after 2020. However, according to the latest surveys, the world is moving further and further away from this goal [6,7].

According to the International Energy Agency's Efficient World Strategy (EWS), at least a 3% improvement would be needed, each year, to meet global climate and sustainability goals. The 3% has only been realized once, in 2015, and the pace has slowed gradually, thereafter.

By using cost-effective technologies [8], the pace of energy efficiency improvement can be greatly increased. Design principles such as modular device design, maintainability and traceability can serve this goal. Digitalization, remote supervision systems and intelligent monitoring are closely linked to these features [9–11]. The continuous development of end-user technological capabilities also supports the need for modular, easy-to-maintain

designs [12]. There is a niche market for manufacturers to ensure user-friendliness and repairability of civil and industrial devices—in exchange for some extra cost. The authors hope that the modular design and the installation and repair manuals of the devices might again be common practice.

In terms of robustness of such systems, successful results have been achieved in the field of fault diagnosis and fault tolerant techniques [13–15]. Basically, we can define the following categories for fault diagnosis methods:

- Model-based [16–21]: the outputs of the system-model and the outputs of the real system are compared to each other.
- Signal-based [22–28]: a diagnostic decision is made based on the measured signal.
- Knowledge-based [29–33]: evaluation is based on a large volume of historic data.
- Hybrid and active [34–38]: combination of the previous methods based on their advantages.

In the literature, several practical solutions can be found for robust electronic circuits [39–42]. In the field of off-grid power supply, a common approach to increase reliability and redundancy is to use hybrid systems, which could include for example photovoltaics, wind turbines or diesel generators [43]. Moretti et al. propose a predictive control strategy for photovoltaic and dispatchable generator hybrid systems, resulting in a 6.5% cut of the overall system cost [44]. Petersen et al. discuss a sizing algorithm for modular and scalable wind integrated hybrid power plants [45]. Chowdhury et al. analyse the costs of the solar-diesel hybrid systems [46]. The focus of this paper is photovoltaics.

Another perspective of redundancy is where instead of different types of power sources, the duplication of system elements are applied. Tuladhar proposes different topologies for optimum energy extraction from solar panels [47]. Liu et al. discuss a fault-tolerant control for single-phase cascade off-grid photovoltaic-storage system [48].

In this paper, we describe a method that significantly increases the redundancy of the photovoltaic system by duplicating the solar cells. The need for this and certain control issues will be analyzed.

## 2. Pragmatical Approach of the Problem

A centralized network is not a prerequisite for the current energy system. Professionals are placing increasing emphasis on the development of renewable energy sources. As a result of the developments, solar energy solutions are already available for civilian use, which are able to provide electricity to buildings or equipment. This article does not cover more powerful industrial solar systems or concentrated solar energy utilization, due to the areas of application the article focuses on.

Systems that produce electricity from solar energy can be grouped according to their ability to feed back into the electricity grid. Isolated power supply systems perform their function independently of the electrical network, due to the scope of the article, only systems capable of independent operation are discussed.

The system stores the energy produced by the solar cells in the batteries. Energy from the power source or stored in the battery will be used by equipment connected to the power system. The circuit used to charge the battery is responsible for matching the voltage levels of the solar panels to the battery. In this case, it is necessary to take into account the optimization of the load on the solar cell as well as the charging methods that suit the type of battery. Batteries manufactured with different technologies have different charge diagrams. Failure to do so can result in a significant reduction in lifetime.

The function of the load drive circuit is to match the voltage of the battery to the input voltage of the load. A DC/DC converter is required for DC output and an inverter for AC output. For isolated power supply and inverter output stage, synchronization of the mains 50 Hz frequency and the inverter output frequency is not required. For the inverter output voltage waveform, a purely sinusoidal waveform is strongly recommended to increase the lifetime of the load.

The power supply system is powered by solar panels. The output of solar panels can vary widely depending on the lighting (time of day, season, cloudiness or pollution). The purpose of the input circuit is to make the best use of the solar cells in the name of cost-effective installation, for which the solar cells must be matched to the stage that loads them (see Figure 1). For maximum efficiency, power matching of solar panels is required. The task of the battery charging electronics is to set the operating point of the solar panels using the MPPT (Max Power Point Tracking) charging algorithm. The battery charging electronics must be able to measure and handle voltage and current values that vary over a wide range.

Proper orientation of photovoltaic panels is essential for the efficiency of the power generation. Hungary is located at 47° north latitude. The most effective positioning is when the rays of the Sun and the solar panel are perpendicular, thus the degree of reflection is the lowest. It is recommended to set the angle of inclination of the solar panels at an angle of approximately 45° to the south, in the middle position of the rotating mechanism—in this case, the optimal annual energy production can be ensured. The recommended tilt angle is just a guideline for the entire year, as the height of the Sun's orbit is different during the summer and winter. This means a near-horizontal situation in summer at noon, but it would be the most unfavorable position in winter. Compared to the horizontal, it would be advisable to set the angle of the solar panels to 36° in summer and 59° in winter. Another important aspect of the 45° tilt angle is that the snow deposited on the panels will only be removed if the slope is correct; therefore, a tilt angle of less than 40° is not recommended.

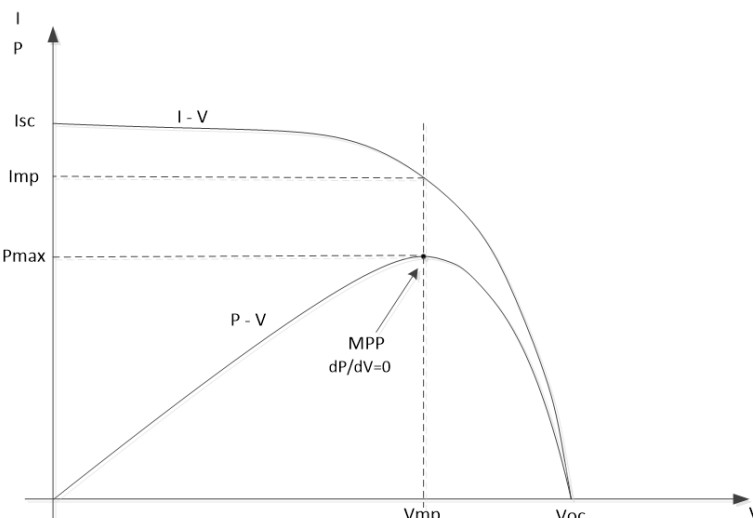

**Figure 1.** Modular power supply structure.

The solution discussed in this article also seeks to optimize costs and system efficiency, as well as to implement a robust mechanical design, so it uses only axial rotation.

Regarding the placement of the equipment, placing on the ground would not be adequate due to the undergrowth and the crown of any trees, so it is recommended equipment be placed at a higher position. It would be more appropriate for a camera system as well if it can observe the terrain from a height. In a wooded area, the appropriate mast height would be 10–15 m.

The maximum efficiency is also affected by the production technology of the selected solar cell. A crystalline silicon solar cell is a commercially available affordable solar cell technology with one of the highest efficiencies (20–24%).

The monitoring system, with its video recording system, wireless connection, and LED that emits infrared light at night, has an average power consumption of 20 W over a 24-hour period, which is 480 Wh of energy in 24 h. Using 12 V batteries and system voltage, this is 40 Ah of charge. Additional adjustments are required depending on the battery technology used, the ambient temperature, and the number of hours of low sunlight expected.

In the case of batteries, the difference between the maximum charge voltage and the minimum discharge voltage must also be taken into account when determining the amount of charge that can be extracted from them. It is recommended that acid batteries with a fixed electrolyte manufactured for industrial use be installed. Quality types can be discharged up to 20% of their rated voltage without damage. Using this technology, the voltage step can be 9.6 V, which means a battery capacity of at least 50 Ah.

Taking into account 30 years of meteorological data, the number of sunny hours in the summer months in Hungary is roughly 3.7 times higher than the hours in the winter months. At an ambient temperature of $-10\,°C$ in winter, the capacity of the battery proposed drops to roughly 65%. These operating conditions assume the worst application environment. Depending on the need to ensure availability, the extent of the increase in battery capacity can be determined, which will inevitably lead to an increase in costs.

Modular systems provide an opportunity for quality testing of the modules that make up the system. The monitoring the outputs and inputs of the modules that make up the system and the self-testing are performed by a microcontroller-controlled hybrid electronic circuit. Based on the measurements, the available capability of the system can be extended by replacing the redundant modules. The microcontroller unit (MCU) transmits the locally determined system information—the results of the procedure for evaluating the modules—over a wireless communication channel to a remote monitoring system. Depending on the configuration, it is possible to use the wireless communication channel of the powered system. In this case, it is necessary to ensure communication between the power supply system and the powered system.

## 3. System Level Approach of Redundancy

For a single-redundant power supply unit (PSU), if one of the PSUs fails, the backup PSU takes over the task (see Figure 2) [49]. This means that only one PSU is working at a time and that it supplies 100% of the required electricity for the powered system. In this case, the backup power supply is out of service or under test. This design ensures that the power supply is fault tolerant. This mode is also called hot-stand-by mode.

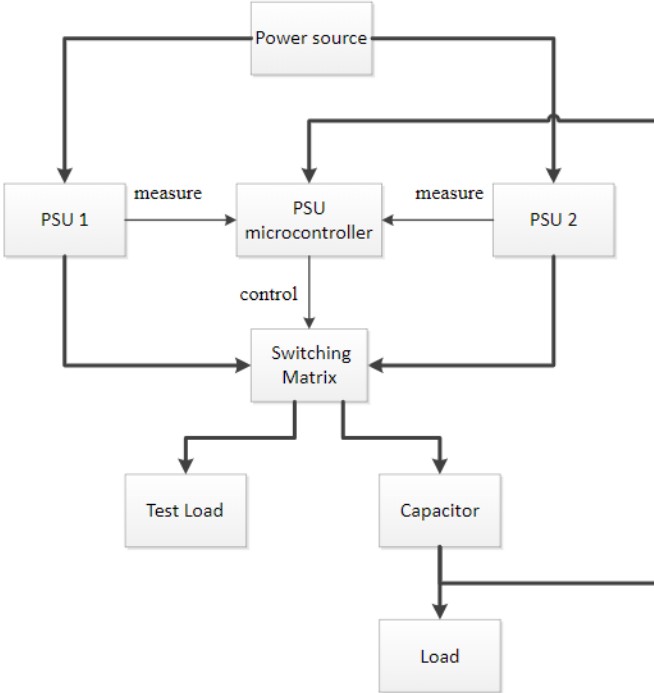

**Figure 2.** Redundant power supply structure with supervisor MCU.

Another solution is the load-sharing mode, where power supplies share the load power. If there are more than two power supplies in the system and one is out of service for failure, replacement, or testing, the remaining power supplies will share the total load current equally.

For example, if there are four redundant PSUs in the power supply system and one of them, for the abovementioned reasons, goes out of service, the power supplies that are still in operation will distribute the load, so for example, the single units should provide 33% instead of 25% of the load current. All power supplies should be able to provide full load current if left alone in the power supply system.

The capacitor supplies power to the microcontroller, which supervises the PSU, the connected load, and other optional modules, when the power supply is temporarily interrupted due to replacement of the redundant modules.

## 4. Electrical Implementation of the Redundant System

Power supplies usually include a microcontroller that affects the power supply and provides measurement, logging, communication functions among others as well. In the case of modular power supplies, the power supply control microcontroller and the tightly-coupled components may be provided as separate modules or may be part of the motherboard (see Figure 3).

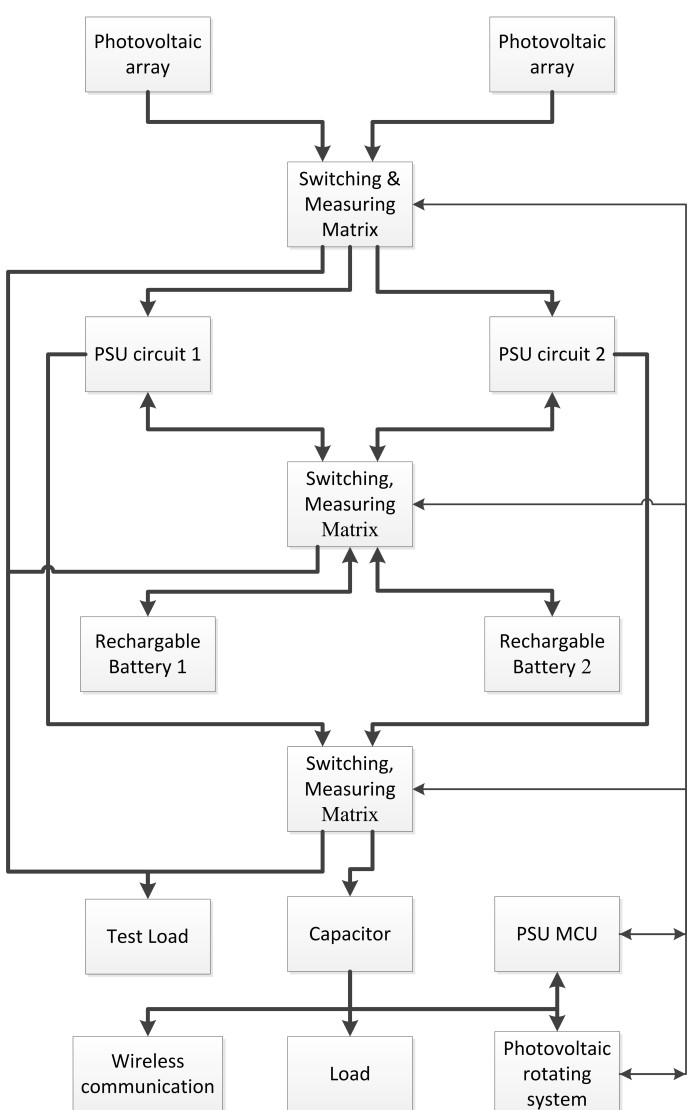

**Figure 3.** Simplified redundant modular power supply unit structure.

For redundant power supplies or redundant modular power supplies, the microcontroller for power supply control extends to monitoring, testing, evaluating the power supplies or power supply modules, and controlling switching matrices.

The microcontroller also controls the switching matrices that connect the power modules (see Figure 4). The function of the switching matrices is to provide energy flow between the power modules in a reconfigurable manner. The switching matrices can be used to connect and disconnect redundant power modules, including replacing redundant power modules [50].

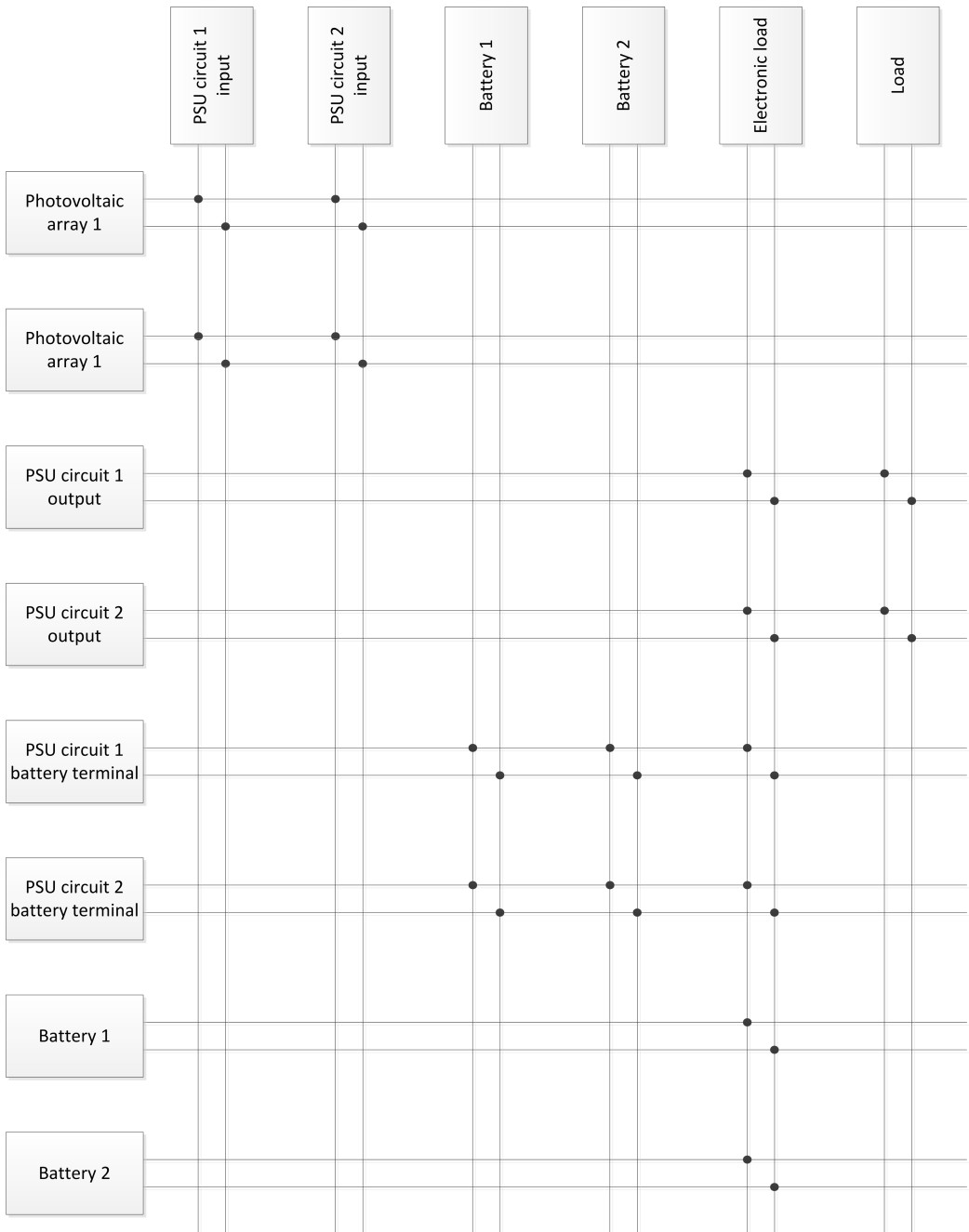

**Figure 4.** Switching matrix structure.

When replacing power modules, switching multiple redundant power modules in a way that would lead to a short circuit should be avoided. The first step is to disconnect the currently active module, after which the redundant module can be turned on. The process results in a short-term power line break, but with the addition of energy storage

devices (buffer capacitors), the power supply is continuously maintained, and transient-low switching is possible. In the experimental setup, galvanically isolated relay modules were used. In case of the end product, it is recommended that modern technology semiconductor switching elements be used.

In redundant fault tolerant systems, some basic features are needed for proper operation. In hot-stand-by mode or load-sharing mode, the embedded monitoring system must be able to notify a supervisor and a control system of the actual status or failure of power supplies and modules.

The embedded monitoring system must be able to monitor power supplies and modules. Depending on the various aspects of error detection, this may occur using a normal operational load or dummy load. Both the normal operation and the test operation must be measured with active and standby power modules as well. This can be done by swapping the power modules or by intermittently disabling (test operation).

Redundant fault tolerant power supply systems must be provided with continuous power supply even when defective modules are replaced. The hot-plug-in function ensures smooth operation of the powered system.

During the measurement of the modular power supply, the module's efficiency, temperature, input and output voltage and current values and waveforms must be monitored. The measured values should be transmitted to the microcontroller for further processing and storage.

Voltage measurement, if higher than the reference voltage of the analog digital converter of the microcontroller, is carried out by a voltage divider made of high-precision and stable elements. There is even the possibility of using an optocoupler, but the brightness degradation of its built-in semiconductor LED, typically in the infrared range, can over time falsify the measurement.

Current measurement can be accomplished by measuring or calculating the differential voltage across Shunt resistors with lower cost. In order to reduce the number of test cables or to measure higher current values, it is also possible to use hall sensors, which have the disadvantage of higher costs.

For lifetime prediction, the voltage difference of the semiconductor drain-source shown in Figure 5 is also measured when the MOSFET is open. When the MOSFET is closed, the drain-source voltage can easily be higher than the input of the operational amplifier used to measure the voltage difference, so the measurement must be constructed with a galvanically isolated operational amplifier.

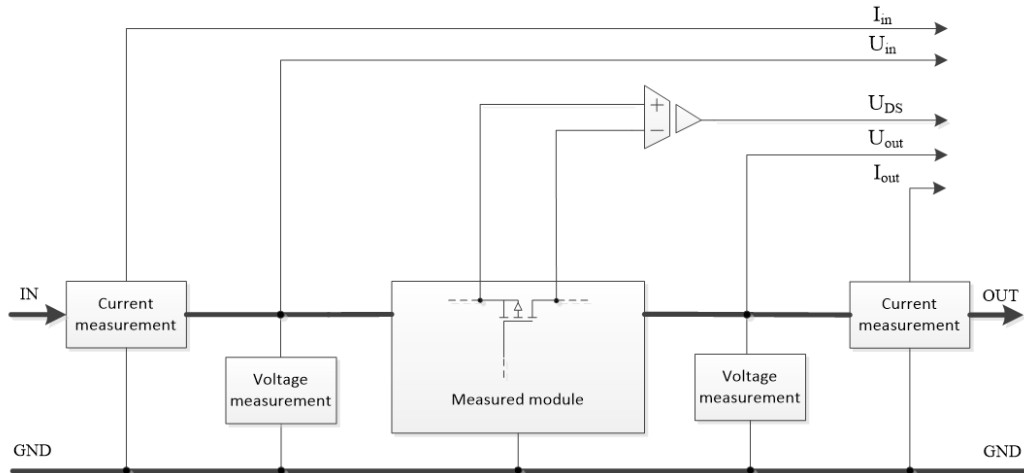

**Figure 5.** Module measurement scheme.

Measurement and evaluation of power modules can be accomplished by using integrated hardware elements which support the measurement. They can also detect overcurrent, overvoltage, voltage drop, voltage fluctuation, instability, voltage loss and other anomalies.

The modules that build up the power supply are connected to each other via the switching matrices, so it is practical to place the subcircuits around the switching matrices, which enable the measurement of electrical parameters (see Figure 6).

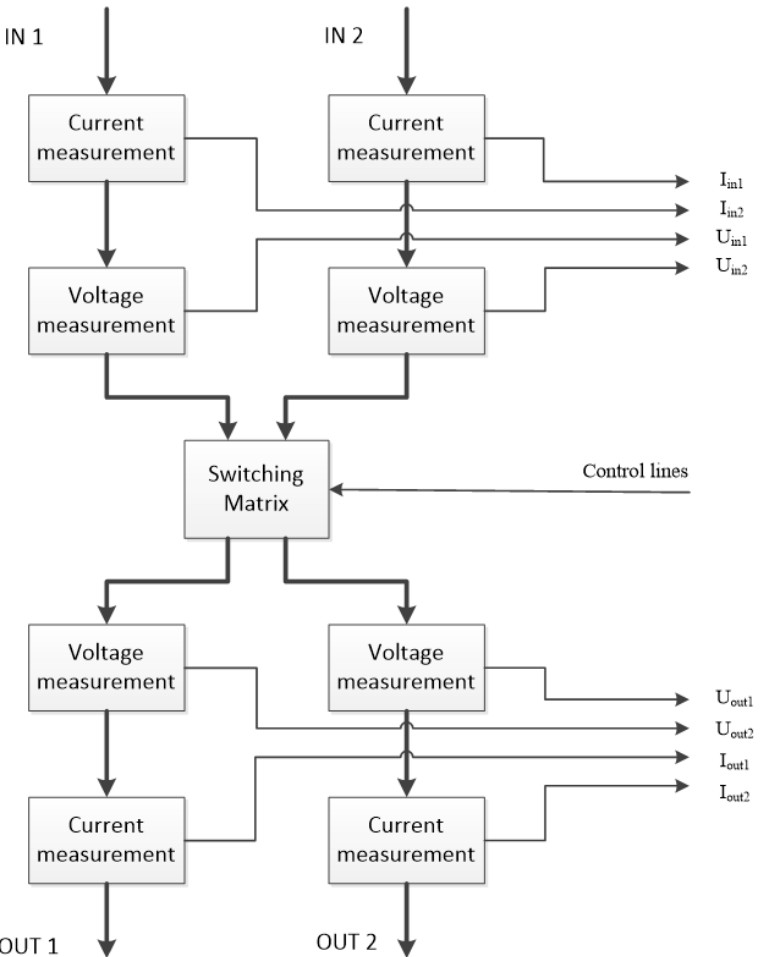

**Figure 6.** Power line measuring concept.

The microcontroller that monitors the power supply and controls the switching matrices has a limited number of terminals and is required to expand due to the large number of measurement and control signals. Extending the number of control outputs is easily accomplished with a serial I/O extender IC. The analog signals to be measured are coupled via an analog multiplexer to the ADC terminals of the microcontroller (see Figure 7).

Choosing a more advanced microcontroller eliminates the need for external hardware, with sufficient software (multiplexing the input terminals to the ADC peripheral) to implement the solution presented. If the microcontroller has multiple internal ADC peripherals, it is recommended to measure the same analog signals with the same ADC modules—this is to avoid measurement errors due to differences in measurement peripherals. If the microcontroller has one internal ADC periphery, a sample and hold (SH) circuit is necessary to be used.

Most of the time, the microcontroller controlling the power supply is in sleep mode. During sleep mode or when executing an instruction, you may not be able to detect unexpected voltage fluctuations in the power supply. Due to the architecture of the interrupt request system, which is designed as an external hybrid circuit, it is able to continuously monitor the output voltage state and, for example, to request an interrupt from the microcontroller in case of voltage fluctuation outside the specified limit.

Unlike the block diagram shown in Figure 8, the use of a built-in comparator as the internal periphery of the microcontroller controlling the power supply is recommended, in which case, the reference voltage can be set as a register content by software. The internal comparator peripheral of the microcontroller also has the ability to request an interrupt.

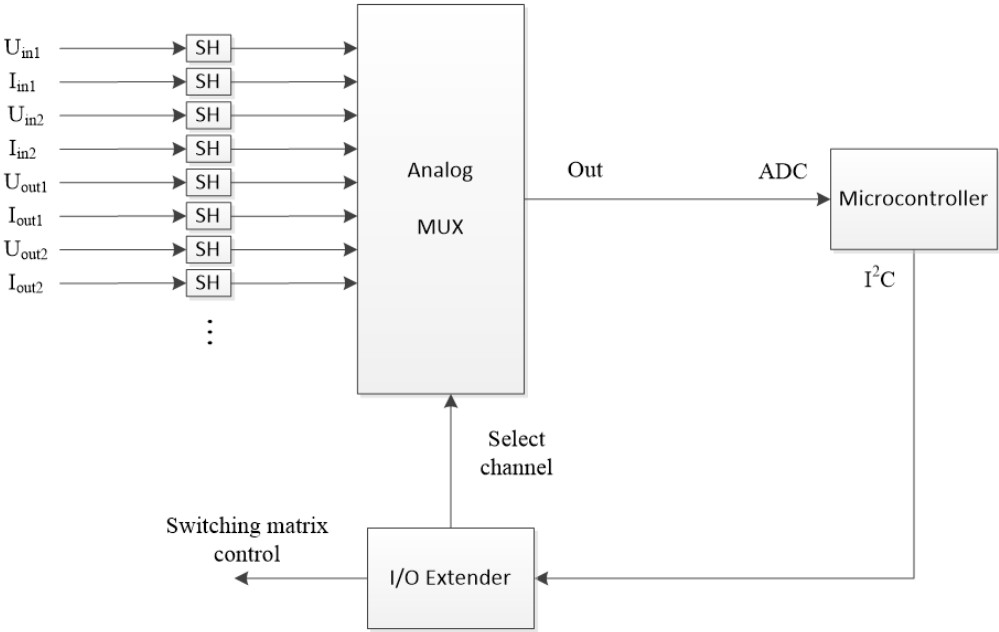

**Figure 7.** Hardware measurement and control scheme.

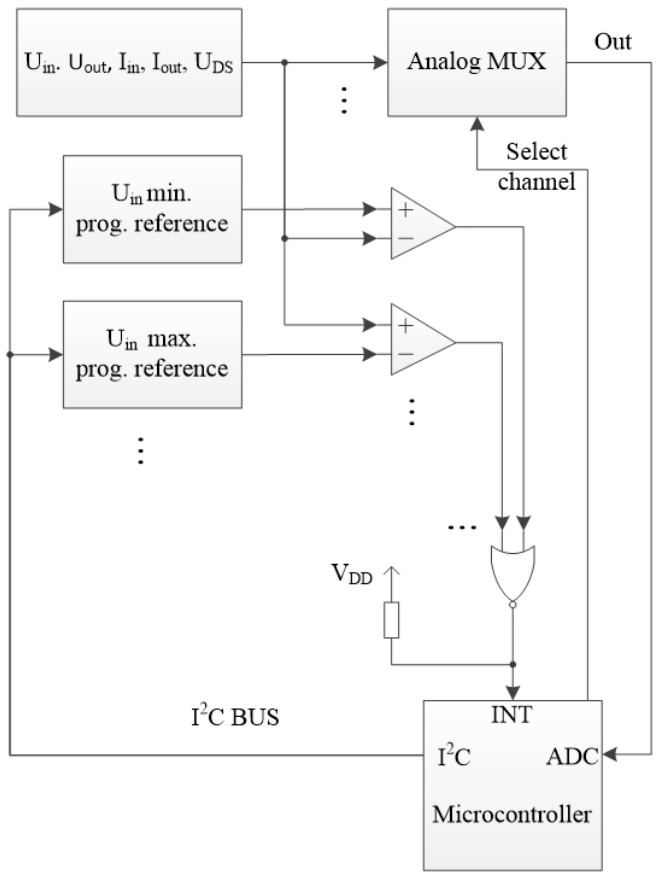

**Figure 8.** Block diagram of the external interrupt system.

For testing the system, a modular power supply system with single redundancy was implemented, which can operate in an isolated manner (see Figure 9 and Table 1). The reduction in the efficiency of the power modules was emulated by serially inserted low-value resistors.

In the course of the development, the authors have been working on data acquisition devices. In an end product, the application of an analogue multiplexer is recommended to extend the number of terminals on the microcontroller.

The determination of the quality of batteries is based on the calculation of efficiency. The change in the ratio of the amount of charge during charging and discharging the batteries correlates with the lifetime of the battery. Steep efficiency declines—which can be caused, for example, by increased internal resistance—predict a future malfunction.

The Li-Po rechargeable module used in this model not only charges the battery, but also protects the batteries. The charging current of the module is equal to the maximum charging current of the batteries. The protection function of the module prevents the batteries from being deep discharged or the development of high load current. As the battery ages, the battery's storage capacity decreases, and its internal resistance increases—performance can be monitored to track battery lifetime.

The quality of the DC/DC converter, which produces the output voltage of the modular power system, is also measured by efficiency. The desired indicator value can be obtained by dividing the input and output power. The converter operates in boost mode and its load capacity exceeds the load current that the batteries can provide, or the load requires.

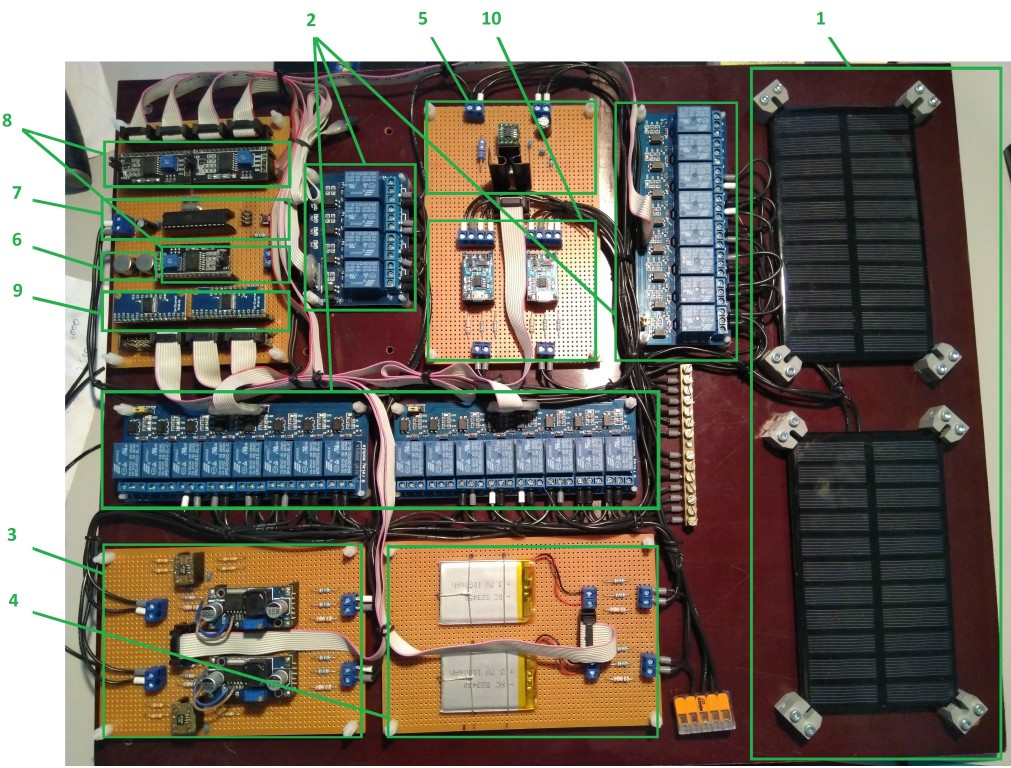

**Figure 9.** Experimental setup.

Galvanically isolated relay modules were used in the experimental model circuit. The built-in visual indicators also helped the development process. In the case of the end product, it is recommended that semiconductor switching elements, MOSFETs, be used as they are more reliable, easier to use with higher switching speed, consume less power, are smaller size, and have a lower price.

**Table 1.** Elements of the experimental setup.

| Nr. | Element |
| --- | --- |
| 1 | Photovoltaic array |
| 2 | Switching and measuring matrix |
| 3 | DC-DC converters |
| 4 | Rechargable batteries |
| 5 | Test load |
| 6 | Capacitor |
| 7 | PSU MCU |
| 8 | I/O extender |
| 9 | Analog multiplexer |
| 10 | Battery chargers |

Like all components, semiconductor switching elements can also fail, which is most likely to occur at MOSFETs under load. If the semiconductor switching element fails, the channel may short-circuit or break. In the first case, the MOSFET cannot be closed (cannot turn off)—it remains open independently of the control—and in the second case it cannot be closed (can not turn on),it remains open independently of the control. Both events can cause the system to malfunction.

In order to increase the fault tolerance of the switching matrix, it is possible to use an increased number of switching elements (see Figure 10). Depending on the level of redundancy, the likelihood of certain types of errors occurring can be significantly reduced. The MOSFETs of the redundant switching unit are controlled in parallel.

If you use two serially connected MOSFETs instead of one MOSFET and one is shorted (no matter which one), the switching unit will remain controllable—the error-free MOSFET can still turn the line on and off. If one of the MOSFETs connected in series fails with a break, the line can no longer be turned on—regardless of the open or closed state of the other MOSFET.

If two MOSFETs are connected in parallel and one fails with a break (no matter which one), the switching unit remains controllable—the error-free MOSFET can still turn the line on and off. If one of the parallel MOSFETs is short-circuited, the line can no longer be turned off—regardless of the other MOSFET's open or closed state.

When two serially connected branches are deployed in parallel, employing a total of four MOSFETs, the coupling becomes capable of repairing both short-circuit and open-type failures at once.

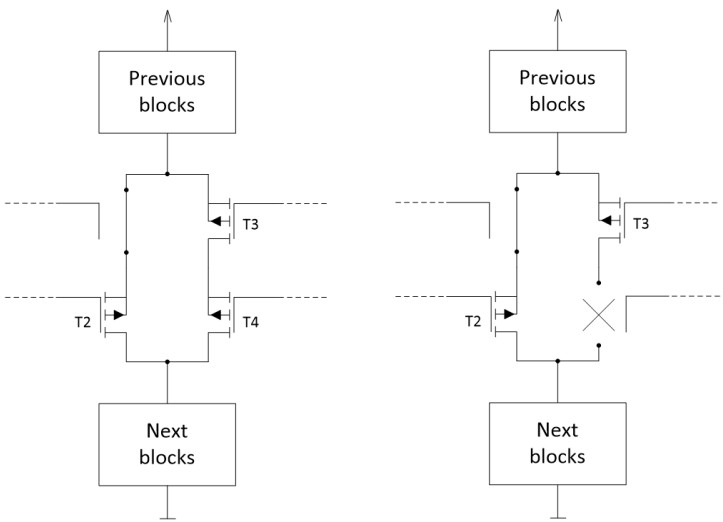

**Figure 10.** Fault-tolerant switching element with redundant MOSFETs.

Controlling power MOSFETs in parallel requires even channel current sharing. The use of MOSFETs in parallel is facilitated by the relatively small differences between their parameters. It is preferred that the gate-source voltage ($U_{GS}$) and the temperature of the MOSFETs be the same for all parallel-connected transistors so that approximately the same current flows through them. In contrast, gates are not recommended to be connected directly to each other, because capacitive inputs and scattered inductances can cause unwanted high-frequency vibrations. In order to avoid high frequency interference, it is advisable to connect the gate to the common control point in series through a low-value damping resistor.

A programmatically variable load is used to test the modules that build up the power supplies. With its application, the power supply modules can be loaded with the same and constant resistance and measured periodically. The switching matrix switches the load on the power supply modules.

In Figure 11, the +5 V branch supplies the circuit and the V+ branch is the voltage source to be loaded. The load can be controlled by changing the channel resistance of the N-channel MOSFET Q2. The load current is proportional to the voltage across the resistor R4. The microcontroller controls the circuit via a PWM signal, the variable duty cycle of the quadrature signal is smoothed by the low pass filter formed by R6, C4. The analog output signal IC1A controls the degree of opening of Q2. The load current at the output of the power supply module to be tested is proportional to the duty cycle.

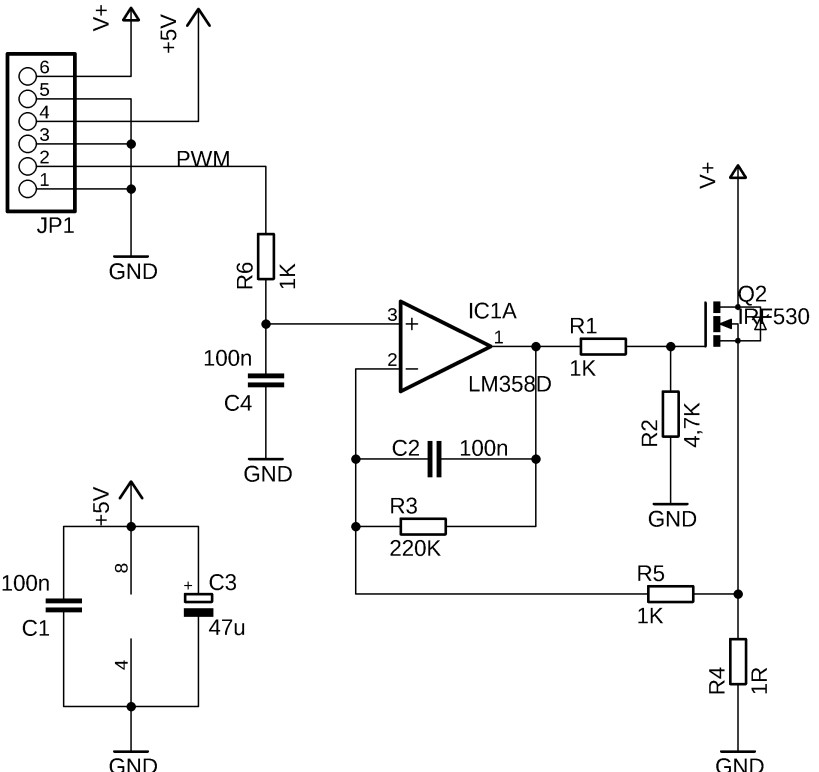

**Figure 11.** Electronic load circuit.

The most common source of power supply failures is a switching or a control element—usually a MOSFET [28]. This may be a stand-alone component or the part of a component with a higher degree of integration. In power MOSFETs, switching can be very fast, thus achieving a low switching loss but not completely omitting it. Typically, the loss of power that is converted to heat on an element can be calculated from the open

drain-source channel resistance ($R_{DS(on)}$) and the current flowing through the element. The main disadvantage is the strong positive temperature coefficient of $R_{DS(on)}$.

$$T_{ep(on))} \approx 6...9 \cdot 10^{-3} \left[ \frac{1}{°C} \right] \tag{1}$$

where $T_{ep(on))}$ is the temperature coefficient of the epitaxial layer. (The load current has no significant effect on the resistance of the open drain-source channel.) The temperature has a significant effect on the resistance of the open drain-source channel. Due to the $\Delta T$ rise in temperature

$$R_{DS(on)} = R_{DS(on)25} \cdot \left( 1 + \Delta T \cdot T_{ep(on))} \right) [\Omega] \tag{2}$$

change occurs, where $R_{DS(on)25}$ is the open drain-source channel resistor at 25 °C. If the temperature of the semiconductor layer increases by 100 °C, the resistance $R_{DS(on)25}$ increases by 1.5 times. For other types of MOSFETs, this can be doubled.

$$R_{DS(on)125} \approx 1.5...2 \cdot R_{DS(on)25} [\Omega] \tag{3}$$

where $R_{DS(on)125}$ is the open drain-source channel resistor at 125 °C. The power loss ($p(t)$) is given by the following:

$$p(t) = i_D^2 \cdot R_{DS(on)} [W] \tag{4}$$

where $i_D^2$ is the drain current.

The resistance of the open drain-source channel increases as the semiconductor layer temperature increases, as does the power loss on the device, and the efficiency of the module decreases.

## 5. Software Solutions

The monitoring system calculates the primary and secondary power of the modules by measuring the input and output voltages and currents of the power supply modules. Based on these measurements, it calculates the efficiency of the power supply modules. If this value is below a predetermined level, or based on several measurements, it can be determined from the stored results that the condition of the unit is deteriorating (the efficiency value drops below a certain level), the monitoring system will send an error message to the monitoring system and jumps to a subroutine.

The monitoring system must process a significant amount of data. The cost-effective microcontroller-observed monitoring system can measure only one point at a time, and the analog-to-digital conversion takes a finitely long time. Power modules have a relatively large number of measurement points. The frequency of each measurement and the accuracy of analog-to-digital conversion (measurement time) can be dynamically changed for efficient operation.

In case the error of a measurement point shows only slight deviation from the ideal, it can be checked at a lower frequency and with less accuracy, so monitoring the measurement point takes only a small amount of time in the cycle (see Figure 12). If the error of the measuring point increases, for example on the basis of a look-up table, it is recommended to increase the frequency and the accuracy of the measurement. If the rate of change of the error at the measurement point increases, it is recommended to further increase the measurement frequency and accuracy of the measurement point to monitor the change. Measured and stored data can be used to perform fault prediction functions.

The software supports user settings. The user has the ability to select the mode of operation and to weigh the following considerations (see Figure 13).

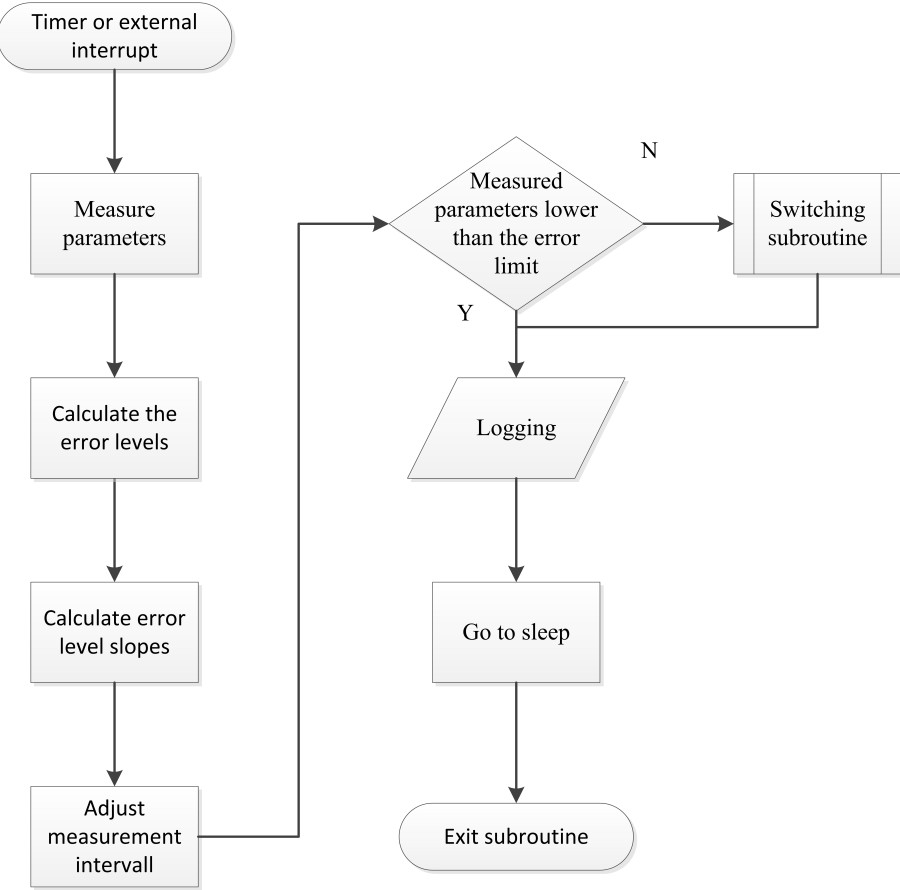

**Figure 12.** Parameter measuring algorithm.

In the case where the user wants to maximize the life of the device—because maintenance is difficult, access to the device is difficult, or the goal is to reduce the carbon footprint—Swapping mode with reduced maximum charging and discharging currents is chosen. The swapping modules subroutine switches between redundant elements primarily based on the efficiency of the modules, but it can also change based on the temperature of the modules (the temperature of the active module increases), thus saving parts from heat stress and faster aging.

With advanced user settings, it is possible to fine-tune the abovementioned parameters, customize reference levels, hysteresis values and timings.

The third mode of operation is the classic Backup mode. The module that builds the primary power supply will operate until it fails, after which the backup module will take over the load. In this case, the backup module may remain reliable for a long time due to higher performance margin compared to the Simultaneously running mode, although it does not include a spare element after the failure of the backup module.

This control mode is used to drive the primary active module until failure or until a predetermined degradation of efficiency, thus the long-term power consumption of this control mode will be the highest.

If higher reliability is the main goal, it is recommended to activate Simultaneously running mode. In this case, the redundant modules run at 50–50% load. The heat load is higher than in the previous case. If one module fails, the other takes over 100% of the load with a smaller transient.

This type of control is recommended for powering easily maintainable equipment. In the case of a module failure, the failure of one of the modules increases the likelihood of a failure of the module remaining in the system, and in the case of the failure of the remaining 100% load module, there are no more spare modules in the system. The 100%

load should only be tolerated by the module remaining in the system until maintenance, so we may use a lower power margin than in Swapping or Backup mode.

In all three cases, especially in the Backup mode, it is important to periodically test the modules and determine their functionality. Measurements can be made with the programmatically variable load or with relatively short operation of the inactive module—until reaching the operating temperature to perform measurements during normal operation.

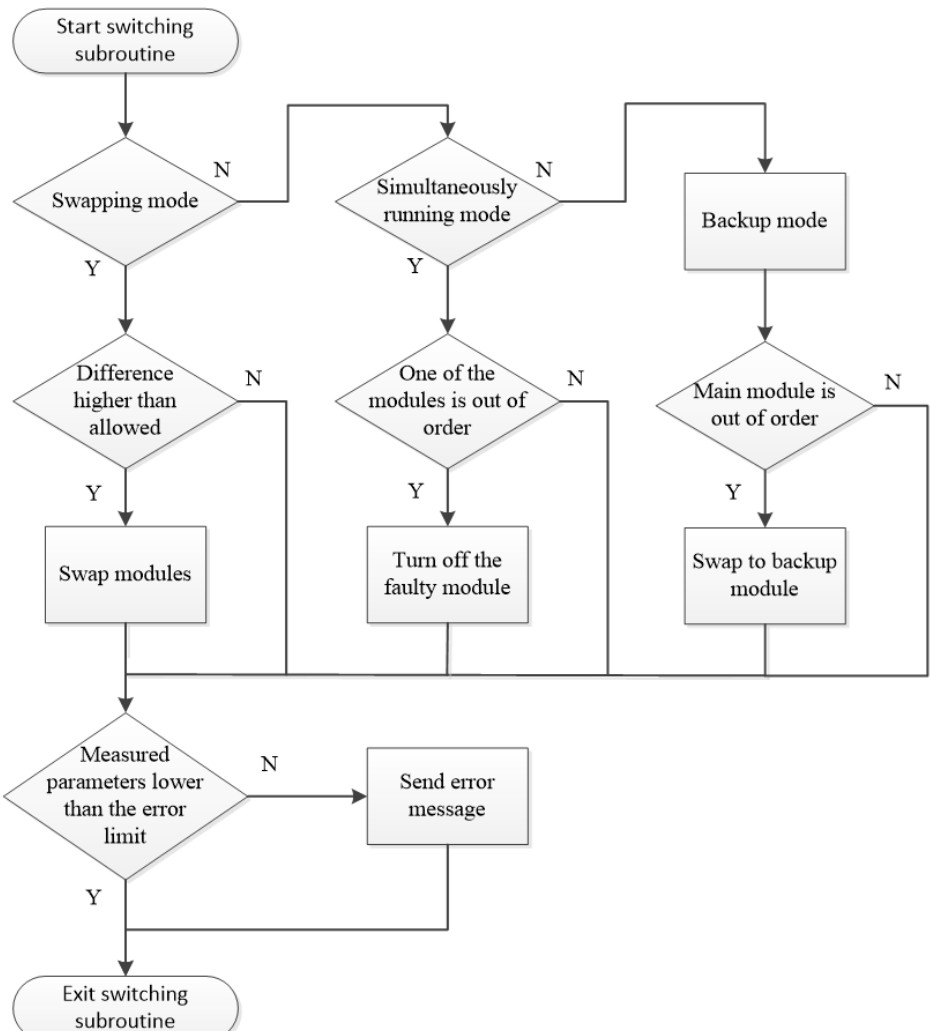

**Figure 13.** Mode selector algorithm.

## 6. Further Possibilities for Enhancing Redundancy

One of the key components of the system is the photo-voltaic component itself, the solar cell. We have already discussed the optimal electronic fit of this, which is a relevant engineering problem. If we analyze the operation of the solar cell, the elimination of its faults is the targeted task. Many articles have discussed the optimal position of the solar cell, following the orbit of the Sun in two axes. Our experimental setup is built based on the literature. The problem taken from practice is the pollution and damage of the solar cells, thus the reduction of the generated energy [51]. Based on user experience, these impurities may be contaminated rain (with dust content), sandstorm (wind carried dust), industrial pollution, dirt from agricultural activities, biology pollution (insects, birds, animals), etc. Injury is an irreversible condition that can be triggered by any mechanical impact, such as hail, human intervention (throwing, shooting). The dirt can be removed with a scheduled or necessary maintenance operation. Mechanical damage can only be repaired by replacement.

It is important to be knowledgeable about performance degradation and it is important to develop some kind of redundant system.

The arrangement shown in Figure 14 is a duplicated uni-axial solar cell. The controlled shaft has a dual function; the solar cell replaces the active and previously inactive solar cell with a substantial 180-degree rotation when the power decreases significantly. The backup solar cell will then probably work with a higher efficiency than what we have declared to be dirty or damaged. A solar cell that has become inactive will be cleaned or replaced during maintenance.

By controlling the axis, we have the opportunity to determine its optimal position relative to the current position of the Sun, constantly adapting. Optionally, a solar sensor semiconductor can be placed next to the solar cell, which can be a great help in the event of a catastrophic failure. An actuator is typically a stepper motor that rotates a mechanism that includes solar panels through a gear or ribbed belt transmission.

It is especially important to develop a solar cell replacement strategy. Comparing the active element to itself should not result in a solution. As an option, if there are several similar solar cells in the system, a comparison with each other should be made. If there is only one duplicate solar cell with the proposed arrangement, then the values of the voltage obtained from the background solar cell, scattered light, can be used for comparison. In the construction according to Figure 14, the unused (i.e., faultless) spare solar cell provides useful measurement data by means of scattered light or artificial lighting. The evaluation that can be implemented using the optimized procedure can be easily performed by interpolating an empirically determined look-up table.

Applying fine rotation of the axis, a simple amplitude to maximum control is performed. The position of the Sun can be detected, for example, by means of electro-optical sensors or by determining the maximum power that can be obtained from the solar cell (see Figure 15). The use of such solutions allows high setting accuracy in good weather conditions. The sensor signals can be processed using analog components (transistors, operational amplifiers) or an analog-to-digital converter.

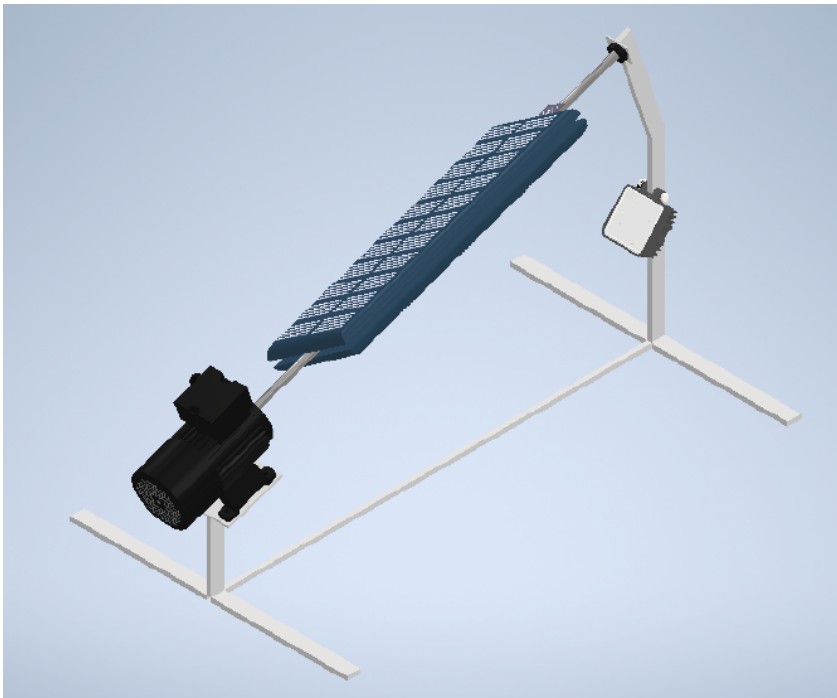

**Figure 14.** 3D model of the duplicated uni-axial solar cell.

U1 and U2 are the two digitized sensor outputs, the solar panel will turn in the direction of the smaller measured value until it reaches the predefined lower error limit resulting from the difference between U1 and U2. In cloudy weather conditions, a cost-

effective determination of the scattered light source is possible based on coordinate and date/time data. The applied microcontroller (or its additional memory) is able to store the chronological data, the increase in the resolution is possible by means of interpolation using the stored data.

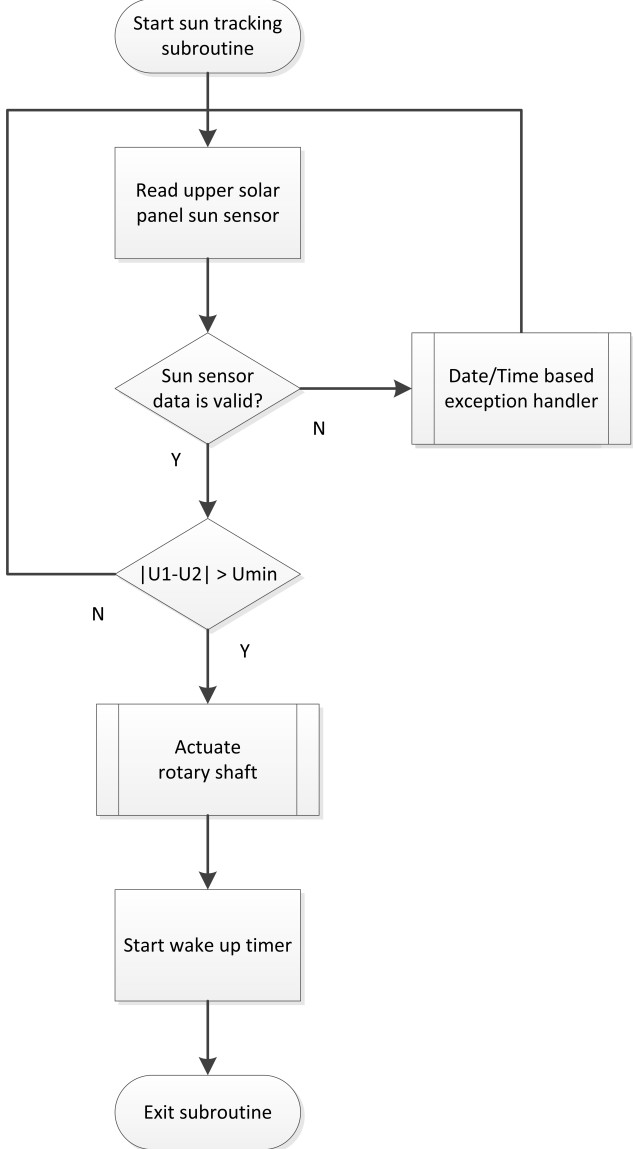

**Figure 15.** Algorithm for maximum voltage control.

## 7. Conclusions

The presented redundant power supply greatly increases the reliability of the device. A cost-effective solution that monitors itself and a modular architecture greatly enhances maintainability, which can significantly increase the life of the equipment, thus reducing global emissions. The duplicated solar cell with a uni-axial rotation mechanism gives the system additional robustness. The authors believe that the presented system architecture can be successfully implemented in both civil and industrial applications, especially for applications requiring high reliability.

**Author Contributions:** Conceptualization, B.B., K.S. and G.G.; methodology, B.B. and G.G.; software, B.B. and G.G.; validation, B.B. and G.G.; formal analysis, B.B. and G.G.; investigation, B.B. and G.G.; resources, B.B., K.S. and G.G.; data curation, B.B. and G.G.; writing—original draft preparation, B.B., K.S. and G.G..; writing—review and editing, B.B., K.S. and G.G.; visualization, B.B., K.S. and G.G.;

supervision, B.B., K.S. and G.G.; project administration, B.B., K.S. and G.G.; funding acquisition, B.B., K.S. and G.G. All authors have read and agreed to the published version of the manuscript.

**Funding:** B.B. thankfully acknowledge the financial support by the ÚNKP-19-3 New National Excellence Program of the Ministry for Innovation and Technology. K.S. thankfully acknowledge the financial support of this work by the Hungarian State and the European Union under the EFOP-3.6.1-16-2016-00010 and GINOP-2.2.1-15-2017-00073 projects. The authors wish to thank the support to the Arconic Foundation and the Howmet Aerospace Foundation.

**Acknowledgments:** The authors thank the community of the Alba Regia Technical Faculty, Óbuda University, for their collegial help.

**Conflicts of Interest:** The authors declare no conflict of interest. The funders had no role in the design of the study; in the collection, analyses, or interpretation of data; in the writing of the manuscript, or in the decision to publish the results.

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
