# Peer review of "Redundant Photo-Voltaic Power Cell in a Highly Reliable System"

_electronics, doi:10.3390/electronics10111253_

Round 1

Reviewer 1 Report

The paper is well presented and the idea as well but in have some questions:

1* The authors should improve the state of the art.

2* There are some references are missing in the list (9 and 10 ..)

3* The authors should present more experimental results and comparative studies to confirm the main objective of the proposed work. 

Author Response

Dear Reviewer,

we would like to express our gratitude for the detailed review and for the helpful comments. Please let us hereby reply to each point by stating how we address them in our revision.

1* The authors should improve the state of the art.
Response: The state of the art have been improved at the end of the Introduction.

2* There are some references are missing in the list (9 and 10 ..)
Response: The references has been updated and corrected.

3* The authors should present more experimental results and comparative studies to confirm the main objective of the proposed work.
Response: The paper describes a phase of development that is the integration of partial results reported in previous publications. It is the outline of a conceptual structure that we plan to implement as a next step (in line 8 of the Abstract the word "implementation" has been changed to "conceptual structure" so as not to mislead the reader). Measurements and evaluations are now being carried out. These practical results are not intended to be part of this publication. 

Yours sincerely,
The Authors

Reviewer 2 Report

The paper concerns a very important topic that concerns the usage of solar modules by off-grid users. The paper is correct and, after small changes, can be printed.

 The author states (page 2): "The output of solar panels can vary widely
depending on the lighting (time of day, season, cloudiness or pollution)." So, it seems that it should be mentioned the possibly ways of improvement reliability of such systems, for instance by combining it with a wind turbine, the mor so as the reliability of the proposed solution is discussed by the authors in detail. Such short discussion can be put forward in Section 6.

In general, the paper is well written. However, there are a few crucial aspects that should be at least mentioned. Furthermore, the bibliography should be completed, which is connected to the said problem. Thus:

page 1, suggested change:

is: [2,3]

suggested: [2,3,3a,3b]

  is: Digitalization and remote supervision systems are closely linked to
      these features [7].

suggested: Digitalization, remote supervision systems and intelligent monitoring are closely linked to these features [7,7a,7b].

Suggested completing of the bibliography:

[3a] Armaroli N., Balzani V. (2007), The future of energy supply: challenges
and opportunities. Angewandte Chemie International Edition, vol.46(1-2), 52–66.

[3b] Bielecki et al. (2020), The externalities of energy production in the context of development of clean energy generation, Environmental Science and Pollution Research, vol.27, 11506-11530.

[7a] Bielecki A., Wójcik M. (2017), Hybrid system of ART and RBF neural networks for  online clustering, Applied Soft Computing, vol.58, 1-10.

[7b] Cambron P. et al. (2018), Control chart monitoring of wind turbine generators using the statistical inertia of a wing farm average, vol.116B, 88-98

Suggested minor changes:

Split Fig.1 into two subfigures: (a) V-I, (b) V-P

Author Response

Dear Reviewer,

we would like to express our gratitude for the detailed review and for the helpful comments. Please let us hereby reply to each point by stating how we address them in our revision.

Combination with a wind turbine.
Response: The suggested hybrid solution has been added to the bibliography and to the Introduction.

The bibliography should be completed.
Response: The suggested changes have been made in the paper.

Split Fig.1 into two subfigures: (a) V-I, (b) V-P
Response: We have followed the common practice and find this representation a better illustration of our intentions.

Yours sincerely,
The Authors

Round 2

Reviewer 1 Report

I would like to thank the authors for the efforts to clarify and addressing my comments, but I still insist on some results that could improve the main of the paper.